# Respiration, Production, and Growth Efficiency of Marine Pelagic Fungal Isolates

**DOI:** 10.3390/jof9040417

**Published:** 2023-03-28

**Authors:** Marilena Heitger, Federico Baltar

**Affiliations:** Department of Functional and Evolutionary Ecology, University of Vienna, 1030 Vienna, Austria

**Keywords:** marine fungi, respiration, production, growth efficiency

## Abstract

Despite recent studies suggesting that marine fungi are ubiquitous in oceanic systems and involved in organic matter degradation, their role in the carbon cycle of the oceans is still not characterized and fungal respiration and production are understudied. This study focused on determining fungal growth efficiencies and its susceptibility to temperature differences and nutrient concentration. Hence, respiration and biomass production of three fungal isolates (*Rhodotorula mucilaginosa*, *Rhodotorula sphaerocarpa*, *Sakaguchia dacryoidea*) were measured in laboratory experiments at two temperatures and two nutrient concentrations. We found that fungal respiration and production rates differed among species, temperature, and nutrient concentration. Fungal respiration and production were higher at higher temperatures, but higher fungal growth efficiencies were observed at lower temperatures. Nutrient concentration affected fungal respiration, production, and growth efficiency, but its influence differed among species. Altogether, this study provides the first growth efficiency estimates of pelagic fungi, providing novel insights into the role of fungi as source/sink of carbon during organic matter remineralization. Further research is now needed to unravel the role of pelagic fungi in the marine carbon cycle, a topic that gains even more importance in times of increasing CO_2_ concentrations and global warming.

## 1. Introduction

The global carbon cycle is driven by many biological processes on land and in the oceans [1], and the interconversions of organic and inorganic carbon compounds are at the core of fluxes and exchanges of carbon between the ocean and atmosphere [2]. The most important biological processes controlling this interplay are photosynthesis—which converts carbon dioxide (CO_2_) to organic carbon, and heterotrophic respiration—which converts the organic carbon back into CO_2_ [3]. The latter is often hard to quantify, but generally, bacterial and zooplankton respiration are depicted as major carbon sinks [4].

Since the introduction of the microbial loop hypothesis [5], it has been widely accepted that microbes are the key players of the processes governing the ocean’s carbon cycle: heterotrophic microorganisms produce new carbon biomass while growing on dissolved organic carbon, and simultaneously consume oxygen and respire CO_2_. A means to relate these two processes is the growth efficiency, which determines how much biomass is produced per unit of assimilated carbon substrate [6]. The current warming of the oceans and the coupled deoxygenation and acidification stresses even more the importance of investigating microbial respiration as a crucial factor balancing the carbon storage in the ocean. Nevertheless, its understanding is still limited by methodology and sufficient surveys, since in-situ respiration is difficult to measure accurately [6,7].

Whereas the role of marine bacteria has been studied quite intensively, the diversity and metabolic potential of aquatic fungi have been largely overlooked [8]. Even though fungi are found in nearly every marine habitat, their contribution to the open ocean food web and carbon cycle has gathered scientific interest only in the past decade [9]. Several studies suggest that marine fungal communities exhibit a high diversity and dominate the biomass on marine snow particles in the bathypelagic realm of the ocean [10,11]. Most free-living planktonic fungi (i.e., mycoplankton) belong to the divisions of Ascomycota and Basidiomycota in the subkingdom of Dikarya [11,12,13], whereas Chytridiomycota have been described as parasites on phytoplankton in productive coastal ecosystems [14]. Planktonic Ascomycota and Basidiomycota species have been found to be present in phytoplankton blooms, contributing to the degradation of the algal polysaccharides by utilizing extracellular enzymes [15]. Recent observations in the North Sea underlined the active role of Dikarya in spring phytoplankton blooms [16]. Moreover, the undeniable yet still largely unexplored role of pelagic fungi in the global carbon cycle has been highlighted in recent global-ocean multi-omics analyses of all carbohydrate-active enzymes [17] and proteins [18], revealing an active role of pelagic fungi in the degradation of carbohydrates and proteins. The findings are supported by another recent study investigating the degradation activity of mycoplankton throughout the water column based on metatransciptome data from the Tara Oceans expedition [19]. The carbon assimilation of marine fungi in the surface as well as the deep ocean and the subseafloor has recently been addressed by combining stable isotope probing and phylogenetic analyses, revealing relevant functional mechanisms structuring the carbon flow in marine ecosystems [20].

Irrespective of their significant influence on the marine carbon cycle, the fungal contribution to microbial respiration has not been yet quantified [21]. So far, there is only one laboratory study available investigating the respiration of different pelagic fungal species isolated off the coast of Chile [22]. Moreover, there is, to our knowledge, no data on production rates or growth efficiency of marine fungi published. Their reported presence in phytoplankton blooms [15,16], however, stresses that these data would be crucial to understand if fungal respiration rather acts as carbon sink or source.

Hence, this study investigates respiration and production rates, and the growth efficiency of marine pelagic fungi. To account for the effect of temperature on the metabolic rates of pelagic fungi, we calculated the Q_10_ factor, which is the change of a metabolic rate with a temperature change of 10 °C [23]. In the ocean, where temperature shifts occur at large spatial and temporal scales [24], the response of organisms and metabolic processes is of special interest in the light of global warming. In parallel, we determined the impact of nutrient concentration on the growth, production, respiration, and growth efficiency of the isolates. Finally, by comparing the metabolic rates and their responses to warming of different fungal strains, we can also shed light on whether changes in fungal community composition might impact community activities and services.

## 2. Materials and Methods

### 2.1. Cultivation of Fungal Cultures

The fungal isolates *Sakaguchia dacryoidea* and *Rhodotorula sphaerocarpa* were obtained from the Austrian Centre of Biological Resources (ACBR), whereas *Rhodotorula mucilaginosa* was isolated during the “Poseidon” Cruise across the Atlantic Ocean in 2019. Briefly, seawater was transferred with a pipet onto a plate filled with a growth media containing: 1 g L^−1^ glucose, 1 g L^−1^ peptone, 1 g L^−1^ yeast extract, 1 g L^−1^ starch, 20 g L^−1^ artificial sea salts, 15 g L^−1^ agar, and 0.5 g L^−1^ Chloramphenicol to avoid bacterial contamination.

The isolates were grown in the dark at room temperature on agar plates (ingredients are shown in Table 1); the media was autoclaved, and Chloramphenicol was added to avoid bacterial growth. Part of the fungal biomass was transferred to a new plate with a sterile cell spreader approximately one week before starting the actual experiment in the liquid culture.

The experiment involved two rounds of testing for each species: first, in a media with a high concentration of glucose, yeast extract, and peptone at 5 °C and 15 °C to test differences according to temperature; and second, in a media with low nutrient concentration at 15 °C to observe differences based on nutrient availability (concentrations are shown in Table 1). Both experiments were carried out at a salinity of 30, obtained by adding a pre-prepared sea salt mixture (Sigma Aldrich, St. Louis, MO, USA) in the high nutrient media and by using artificial seawater (ASW) in the low nutrient media.

The stock ASW contained (per L Milli-Q water): 24.54 g NaCl, 4.09 Na_2_SO_4_, 0.7 g KCl, 0.2 g NaHCO_3_, 0.1 g KBr, 0.003 g H_3_BO_3_, 0.003 g NaF, 11.1 g MgCl_2_ · 6H_2_O, 1.54 g CaCl_2_ · 2H_2_O, and 0.017, SrCl_2_ · 6H_2_O, and was diluted to a salinity of 30 with Milli-Q. After adjusting the pH to 8.1, the medium was autoclaved, and 0.5 g L^−1^ Chloramphenicol were added to avoid bacterial contamination.

To prepare the liquid cultures, a sterile cell spreader was used to dilute fungal biomass taken from the agar plates in Milli-Q water. Dilution was carried out until an optical density of 1 was achieved (measured at 660 nm in a spectrophotometer), indicating that the fungal biomass was dispersed in the water and the initial biomass was the same for each experiment and species. Once diluted, the suspension was inoculated to the media in a ratio of 1:100. After inoculation, the culture was distributed to 7 Schott bottles, resulting in aliquots of 200 mL fungal culture in each bottle, which were incubated in a shaking incubator (Jeio Tech ISS-7100 Incubated Shaker, Daejeon, South Chungcheong, Republic of Korea) at 140 rpm and at 5 °C and 15 °C, respectively.

The growth of the cultures was tracked by measuring the optical density (OD) of one aliquot (control bottle) in a spectrophotometer (VWR, Radnor, PA, USA) at a wavelength of 660 nm. Based on OD, the growth phase could be estimated (values are shown in Table 2). During the exponential growth phase, subsamples were taken at two timepoints in triplicate to measure OD, fungal abundance, biomass, and respiration.

### 2.2. Quantification of Fungal Biomass

For collecting the fungal biomass in the cultures, 5 to 20 mL of each triplicate were vacuum filtered through pre-combusted glass microfiber filters (Whatman, 25 mm diameter, 0.7 µm) at −200 mbar. The filters were wrapped in combusted aluminum foil and stored at −20 °C for further analyses. The biomass collected on the filter represents the particulate organic carbon (POC) in the cultures, whereas the dissolved carbon was excluded by the filtration. To estimate the amount of fungal carbon biomass in the culture, the Joint Global Ocean Flux study (JGOFS) protocol for measuring POC was followed [25]. After thawing, the filters were dried with silica gel for 24 h, followed by fumigation in a desiccator for 24 h with 37% hydrochloric acid (HCl) to remove inorganic carbon. Subsequently, the filters were dried again for 24 h with silica gel and afterwards prepared for the analysis in a CHNS elemental analyzer (Vario Micro Cube, Elementar^®^, Langenselbold, Germany) by wrapping them in tin capsules. For each set of triplicates, the adsorption of dissolved organic carbon from the growth media on the filters was determined by measuring the carbon content of only the media, which was then subtracted from the respective triplicates. To estimate fungal production, the mean carbon content (in mg C L^−1^ h^−1^) of the triplicates from the first sampling point was subtracted from the one from the second sampling point and the result was divided by the time in hours between measurements.

### 2.3. Fungal Abundance

Two millilitres of each sample were fixed with glutaraldehyde (2% final concentration) and stored at −80 °C. These samples were used to measure cell abundance in the fungal cultures with a flow cytometer (BD Accuri C6™, Franklin Lakes, NJ, USA). Due to the measurement limit of maximum 1000 cells µL^−1^, the samples were diluted with pre-filtered (0.2 µm) Milli-Q water by a factor of 2, 4, 8, or 50 to a final volume of 500 µL. To stain the cells, 5 µL of SYBR-Green were added to the samples and incubated for 10 min. After that, the cells were counted for 2 min in the flow cytometer, and total cell numbers were estimated using the respective size spectra of the fungal cells (3–5 µm diameter). Each triplicate was measured two times and the mean of these technical replicates was used to calculate the mean and standard deviation of the biological triplicates.

### 2.4. Fungal Respiration

The oxygen concentrations in the same samples were measured via optical sensors. To allow continuous oxygen measurements, optical sensor spots (PreSens Precision Sensing GmbH, Regensburg, Germany) were attached to the inner wall of 120 mL BOD bottles with silicon glue. Before each measurement, a two-point calibration of the sensors was conducted, following the manual of the company [26]. Since the solubility of oxygen is strongly temperature dependent, the calibration was done at 5 °C and 20 °C to stay close to the respective incubation and measurement temperature.

For measuring the oxygen consumption of the fungal cultures, the bottles were filled completely with one triplicate sample each, closed airtight and put into a water bath set to the respective temperature of 5 or 15 °C. After an adaptation period of 10 min, as previously suggested [27], the oxygen concentration was measured every 10 s with an oxygen meter (Fibox 4 trace, PreSens GmbH) that was connected via a glass fiber cable to the optode from outside of the bottle. The oxygen concentration of only the media was measured for around 20 h a priori, to exclude chemical oxygen consumption by the ingredients of the media or contamination. Since the Fibox 4 trace can analyze only one sample at a time, it was required to switch the sensor between the replicates every 20 to 30 min. The samples were kept in the water bath and measured continuously until the oxygen concentration was below 50 µmol L^−1^. The respiration rates were calculated from the slope of the regression line from oxygen decrease over time and a respiratory quotient (RQ) of 1 was applied to convert the final oxygen consumption rates into carbon units using the Formula (1) [7]:RQ = ΔCO_2_/−ΔO_2_(1)

Hereby, the oxygen consumption rate was converted to carbon in µmol C L^−1^ h^−1^.

### 2.5. Calculation of Fungal Growth Efficiency and Q_10_

Fungal growth efficiency (FGE) was calculated using the following Formula (2):FGE = FP/(FP + FR) × 100(2)
where FP is the fungal production and FR fungal respiration. To calculate FGE, the average respiration rate of both sampling points was used to achieve the most accurate results. Q_10_ values were calculated for the rates of respiration, cell-specific respiration, and production by dividing the results at 5 °C and 15 °C in the high nutrient media. For each species, the sampling points with the most similar OD values at 5 °C and 15 °C, respectively, were chosen.

### 2.6. Data Analyses and Statistics

The linear regression for calculating the respiration rates as well as the calculation of production and growth efficiency were conducted in MS Excel 2019 and by using R statistical software v4.2.2 [28]. Statistical analyses to compare respiration and cell-specific respiration rates were performed in RStudio [29]. Normal distribution and homogeneity of variances were tested with Shapiro–Wilk and Bartlett’s tests, respectively. To test for significant differences between the three species, their respiration rates were compared with analyses of variances (ANOVAs), using a significance level of 0.05. When overall significant differences between the species were detected, post-hoc TukeyHSD tests were applied to determine which species differed significantly from each other. When only two species were compared, Student’s *t*-tests were conducted. Following the same procedure, differences between the rates at different temperature or nutrient concentration of each species were investigated. The data was visualized using the packages ggplot2 v3.3.5 [30], gridExtra v2.3 [31] and ggpubr v0.5.0 [32].

## 3. Results

### 3.1. Growth Dynamics of the Fungal Cultures

In the high nutrient media, *Rhodotorula mucilaginosa* was the fastest growing species, followed by *Rhodotorula sphaerocarpa* and *Sakaguchia dacryoidea* at both temperatures (Figure 1). In the low nutrient media, the growth yield was lower; however, all three species reached the stationary phase faster than in the high nutrient media and the exponential phase lasted only a few hours (Figure 2). *Sakaguchia dacryoidea* was the first species to reach the exponential phase, followed by *Rhodotorula mucilaginosa* and *Rhodotorula sphaerocarpa*. At the two sampling timepoints during the exponential growth phase, the optical density of the respective triplicates was measured and is summarized in Table 3.

### 3.2. Fungal Production (FP)

The production rates of all species and both media and temperatures are depicted in Figure 3. During the exponential growth phase in the high nutrient media at 5 °C, *Rhodotorula mucilaginosa* exhibited a biomass production rate of 6.44 mg C L^−1^ h^−1^, *Rhodotorula sphaerocarpa* of 1.68 mg C L^−1^ h^−1^, and *Sakaguchia dacryoidea* of 0.70 mg C L^−1^ h^−1^. At 15 °C, the production rate of *R. sphaerocarpa* was 7.85 mg C L^−1^ h^−1^ and the rate of *S. dacryoidea* was 1.01 mg C L^−1^ h^−1^. The effect of temperature on fungal production was expressed in Q_10_ values of 4.68 (*R. sphaerocarpa*) and 1.46 (*S. dacryoidea*) (Table 4).

In the low nutrient media, the POC concentration in the cultures was lower. The production rate of *R. mucilaginosa* was 0.36 mg C L^−1^ h^−1^, of *R. sphaerocarpa* 0.48 mg C L^−1^ h^−1^, and the highest production rate in the low nutrient media was done by *S. dacryoidea* with 1.22 mg C L^−1^ h^−1^.

### 3.3. Fungal Abundance

The cell abundance of all species at both sampling points (SPs, see Figure 1 and Figure 2 for reference) is summarized in Table 5. The highest increase in abundance between sampling points occurred in *Sakaguchia dacryoidea* in the high nutrient media at 15 °C. As an exception, the abundance of *Rhodotorula sphaerocarpa* decreased between sampling points in the low nutrient media.

### 3.4. Fungal Respiration (FR)

Respiration was measured at the two sampling points during the exponential growth phase, where also the carbon content was measured to determine the production rate (see Figure 1 and Figure 2). The oxygen consumption was almost always higher at the second sampling point, except for few cases where the culture had already reached the early stationary phase. To compare the respiration rates between species and temperatures, the sampling points with the most similar optical density values were chosen.

The highest respiration rate was performed by *Rhodotorula sphaerocarpa* in the high nutrient media at 15 °C at the second sampling point (211.87 ± 9.95 µmol O_2_ L^−1^ h^−1^); the first sampling point exhibited lower respiration with 136.33 ± 27.04 µmol O_2_ L^−1^ h^−1^. This was still higher than the respiration rate of *Sakaguchia dacryoidea* at the second sampling point, which was 111.08 ± 1.72 µmol O_2_ L^−1^ h^−1^, around double the rate at the first sampling point (53.13 ± 4.84 µmol O_2_ L^−1^ h^−1^). The mean oxygen consumption rates of the two species at the second sampling point, which resembled in terms of OD, differed significantly from each other (*T*-test; *p* < 0.01; Figure 4b).

At 5 °C, *Rhodotorula mucilaginosa* exhibited the highest respiration of the three species, which was 93.21 ± 4.53 µmol O_2_ L^−1^ h^−1^ at the second sampling point. The OD, however, was considerably higher than the one of *R. sphaerocarpa* (Figure 1), where the respiration rate was 55.90 ± 4.38 µmol O_2_ L^−1^ h^−1^. The same was observed for *S. dacryoidea*, which exhibited a respiration rate of 47.44 ± 3.80 µmol O_2_ L^−1^ h^−1^ at the second sampling point and 24.78 ± 7.24 µmol O_2_ L^−1^ h^−1^ at the first. *R. sphaerocarpa* exhibited a respiration rate of 19.60 ± 0.93 µmol O_2_ L^−1^ h^−1^ at the first, and 30.38 ± 2.70 µmol O_2_ L^−1^ h^−1^ at the second sampling point. The second sampling point of this species was used for the comparison due to the resembling OD (Figure 1). There was an overall significant difference between the species (ANOVA; *p* < 0.01); however, no significant difference was found between the respiration rates of *R. sphaerocarpa* and *S. dacryoidea* (TukeyHSD post-hoc test; *p* = 0.42; Figure 4a). Temperature affected the respiration rates for both *R. sphaerocarpa* and *S. dacryoidea,* as indicated by Q_10_ values of 4.49 and 2.14, respectively (Table 4).

In the low nutrient media, the oxygen consumption rates were lower compared to the high nutrient media at 15 °C, and in a similar range than at 5 °C. The respiration of the first sampling point of all three species was used for comparison and again, *Rhodotorula mucilaginosa* exhibited the highest respiration rate (40.32 ± 1.63 µmol O_2_ L^−1^ h^−1^), followed by *Rhodotorula sphaerocarpa* (35.36 ± 2.63 µmol O_2_ L^−1^ h^−1^) and *Sakaguchia dacryoidea* (34.47 ± 1.12 µmol O_2_ L^−1^ h^−1^). Again, the respiration rates of the three species differed significantly from each other (ANOVA; *p* < 0.05); however, there were no significant differences between the respiration rates of *R. sphaerocarpa* and *S. dacryoidea* (TukeyHSD post-hoc test; *p* = 0.76; see Figure 4c).

Comparing the respiration rates of *R. sphaerocarpa* and *S. dacryoidea* at the different temperature and nutrient conditions revealed that the values were always significantly higher in the high nutrient media at 15 °C than at 5 °C or in the low nutrient media (ANOVA; *R. sphaerocarpa*: *p* < 0.01; *S. dacryoidea*: *p* < 0.01; Figure 5). However, the post-hoc TukeyHSD test showed that the respiration rates did not differ significantly between the high nutrient media at 5 °C and the low nutrient media (*R. sphaerocarpa*: *p* = 0.58; *S. dacryoidea*: *p* = 0.07).

### 3.5. Cell-Specific Respiration

Combining the respiration rates with the respective fungal abundance revealed the oxygen consumption per fungal cell, which differed only slightly between the two sampling points of each species. Contrary to the total respiration, *Sakaguchia dacryoidea* exhibited the highest cell-specific respiration in the high nutrient media at 5 °C (15.69 ± 2.34 fmol O_2_ cell^−1^ h^−1^, Figure 6a), followed by *Rhodotorula sphaerocarpa* (12.80 ± 4.49 fmol O_2_ cell^−1^ h^−1^) and *Rhodotorula mucilaginosa* (8.77 ± 0.59 fmol O_2_ cell^−1^ h^−1^). The cell-specific respiration did, however, not differ significantly between the species (ANOVA; *p* = 0.073). At 15 °C, however, *R. sphaerocarpa* exhibited higher cell-specific respiration than *S. dacryoidea* (17.07 ± 1.67 fmol O_2_ cell^−1^ h^−1^; 14.09 ± 2.95 fmol O_2_ cell^−1^ h^−1^), but the difference was not significant (*T*-test; *p* = 0.2; Figure 6b).

The Q_10_ of cell-specific respiration was 1.33 for *R. sphaerocarpa* and 0.89 for *S. dacryoidea* (Table 4), indicating that the effect of temperature was not very pronounced for either species.

The overall highest cell-specific respiration was measured in the *S. dacryoidea* culture in the low nutrient media (20.76 ± 1.86 fmol O_2_ cell^−1^ h^−1^), followed by *R. sphaerocarpa* (16.60 ± 3.57 fmol O_2_ cell^−1^ h^−1^), and *R. mucilaginosa* (11.90 ± 0.21 fmol O_2_ cell^−1^ h^−1^); the species differed significantly from each other (ANOVA; *p* < 0.05). Whereas cell-specific respiration rates of *R. sphaerocarpa* showed no significant difference from the other two species, the cell-specific respiration rates of *S. dacryoidea* and *R. mucilaginosa* differed significantly from each other (TukeyHSD post-hoc test; *p* < 0.01; Figure 6c).

The cell-specific respiration rates of *R. sphaerocarpa* did not differ significantly between the different temperature and nutrient conditions (ANOVA; *p* = 0.32; Figure 7a). *S. dacryoidea* exhibited significantly different cell-specific respiration rates among treatments (ANOVA; *p* < 0.05); however, the rate at 5 °C in the high nutrient media was not different than at 15 °C (*p* = 0.71) or the low nutrient media (*p* = 0.09) (Figure 7b).

### 3.6. Fungal Growth Efficiency (FGE)

The highest fungal growth efficiency (FGE) was calculated for *Rhodotorula mucilaginosa* at 5 °C in the high nutrient media, with 87%. It was followed by *Rhodotorula sphaerocarpa* (85%) and *Sakaguchia dacryoidea* (62%). At 15 °C, the growth efficiencies of both *R. sphaerocarpa* and *S. dacryoidea* were lower with 78% and 50%, respectively. In the low nutrient media, *S. dacryoidea* exhibited the highest growth efficiency with 75%. The FGE of *R. sphaerocarpa* was 51% and *R. mucilaginosa* had an FGE of 43%.

## 4. Discussion

All three species cultivated in this study belong to the division of Basidiomycota in the subkingdom Dikarya. This phylogenetic group has been described as the most abundant fungal subkingdom in the open ocean [11,12,13], and a more recent study reported Basidiomycota as the most prevalent fungal phylum during a phytoplankton bloom in the North Sea [16]. Significant differences between the species were found in most of the measured metabolic parameters, which can indicate that the composition of the fungal community may be critical for the community activity. All fungal species exhibited fast growth curves in all experiments, supporting recent findings of marine basidiomycete yeasts behaving as opportunistic r-strategists in phytoplankton blooms [16].

Surprisingly, the growth rate of the cultures (meaning, in this case, the time to reach stationary phase) was not significantly slowed down by nutrient limitation, the exponential growth phase of all three species was even achieved faster and lasted shorter in the low nutrient media. Nevertheless, the growth yield was considerably higher in the high nutrient media. Temperature, on the other hand, slowed down the fungal growth, and production rates remarkably, while the growth yield was comparable at both temperatures in terms of optical density and biomass (POC). The production rates of the *Rhodotorula* species were higher in the high nutrient media, while *Sakaguchia dacryoidea* exhibited a higher production rate at lower nutrient concentrations. To put it into context, the measured FP (0.36 to 7.85 mg C L^−1^ h^−1^) exceeded the production of batch cultures of a natural bacterial community from the Mediterranean, which exhibited only around 0.03 mg C^−1^ L^−1^ h^−1^ in nutrient enriched media [33]. Pure cultures of strains of marine bacteria (*Roseobacter* and *Cytophaga*) expressed production rates of 0.02 to 0.03 mg C^−1^ L^−1^ h^−1^ [34]. The higher production in fungal cultures could, however, also be due to the mere difference in cell size. The Q_10_ of FP was 4.68 (*Rhodotorula sphaerocarpa*) and 1.46 (*Sakaguchia dacryoidea*), which indicates that the production rate of *R. sphaerocarpa* is more temperature-dependent than *S. dacryoidea* and the latter can produce biomass at lower temperatures almost as fast as at higher temperatures. The results are comparable to the Q_10_ of other growth parameters such as the growth rates of prokaryotes (2.1–3.9) [35], and nanoflagellates (2.66) [36].

This study is a first attempt to assess the respiration and growth efficiency of pelagic fungi to investigate their role in the marine carbon cycle. In a previous study where five fungal isolates from the coast off Chile were cultivated under different conditions, respiration rates between 156.4 and 273.1 µmol O_2_ L^−1^ h^−1^ were reported [22]. Even though the glucose concentration in this experiment was ten-fold higher than in the present study, the numbers are in the same range (111.08–211.87 µmol O_2_ L^−1^ h^−1^ in our study at comparable conditions). When the authors lowered the glucose concentrations to 0.1 and 0.01 g L^−1^, respectively, the obtained respiration rates (at 20 °C) were between 65.5 and 108.5 µmol O_2_ L^−1^ h^−1^ [22]. Given the 5 °C difference, the numbers are again in a similar range as the respiration of the basidiomycete yeast species used in this study. The lack of other available studies about marine fungal respiration makes it difficult to compare the numbers, while stressing the importance of research in this field even more. The dependence of respiration rates on the nutrient concentration points towards a considerable loss of CO_2_ fueled by fungal respiration during phytoplankton blooms, where high nutrient concentrations occur over a relatively short time span and fungi are highly abundant [16]. While the bacterial contribution to total oxygen consumption was found to account for up to 79% [37], the extent to which fungi contribute to the respiration of oceanic systems is yet to be discovered. The cell-specific respiration rates determined in this study (7 to 20 fmol cell^−1^ h^−1^) are considerably higher than rates reported for a cultivated bacterial community from the Mediterranean Sea (0.1 to 2 fmol cell^−1^ h^−1^) [33]. Marine bacteria in their natural environment exhibit even lower cell-specific respiration rates of around 0.003 fmol cell^−1^ h^−1^ [38,39]. While bacterial cell-specific respiration seemingly increases with depth [38], the present study suggests that the cell-specific respiration of fungi is not majorly influenced by temperature or nutrient concentrations. An exception is *Sakaguchia dacryoidea,* which exhibited higher rates at lower nutrient concentrations and at the lower incubation temperature, suggesting an adaptation of this species to nutrient-limited conditions as well as lower temperatures. The Q_10_ values of fungal respiration in this study—2.14 (*S. dacryoidea*) and 4.49 (*R. sphaerocarpa*)—point towards an important influence of temperature on fungal respiration. The values from this study agree with a Q_10_ of 3.7 reported for microbial respiration in the mesopelagic zone of the South Atlantic and Indian Ocean [40], while they are slightly higher than—for example—the Q_10_ of marine copepod respiration [41]. In general, respiration is accelerated by rising temperatures to a greater extent than primary production [7,23]. The present results suggest that fungi are no exception, and the effect of temperature can be even more pronounced than for other organisms. This conclusion is also supported by the recent study investigating fungal respiration, where Q_10_ values between 2.2 and 6.8 were reported [22]. In times of global warming, increased respiration could further fuel climate change by increasing carbon losses to the atmosphere.

This study is the first report of growth efficiencies of marine pelagic fungi. Both *Rhodotorula* species exhibited high FGEs with values above 75% in the high nutrient media. In the low nutrient media, both *Rhodotorula* species exhibited lower FGEs, though only *R. sphaerocarpa* can be directly compared due to the temperature. *S. dacryoidea* surprisingly exhibited a higher FGE in the low nutrient media. This adds to the previous suggestion that this species is better adapted to lower nutrient concentrations and temperatures. *R. mucilaginosa* exhibited the highest FGE at 5 °C in the high nutrient media and the lowest FGE in the low nutrient media, indicating that even though this species can grow rapidly in all conditions, the growth efficiency seems to depend on the nutrient conditions. To put these results into context, we compared the fungal growth efficiencies we obtained to bacterial growth efficiencies (BGE) reported for marine cultures. Table 6 reviews previous studies and summarizes growth efficiencies of different experiments. BGEs of around 50% are commonly found in marine bacterial cultures, due to ideal substrate and optimized conditions. The BGE in the global ocean is, however, lower than the one of cultures; in the surface ocean, values up to 20% in coastal and below 5% towards the open ocean were reported [37], whereas in deep water masses, the BGE is normally below 5% [39]. Thus, the FGE of our cultures seem to be in the higher end of BGE. Assuming a similar trend for fungi, the FGE in the open ocean would be lower than the FGE obtained in this study. A general nutrient dependency of FGE was not observed due to the different responses of the species to the changing nutrient concentration. Nevertheless, a substantial FGE in nutrient rich environments, such as phytoplankton blooms, can be expected.

## 5. Conclusions

This study combines two understudied topics: research on marine pelagic fungi, which gathered scientific interest only recently, and respiration, which measurement has always been viewed as difficult. The results suggest that fungal respiration, as well as production and growth efficiency, are high and affected by nutrient concentration, even though the response varies between species. Furthermore, the effect of temperature differences leads to the assumption, that with warming oceans, FR could increase and FGE could decrease, leading to an even higher loss of CO_2_ to the atmosphere. Future studies with more pelagic fungal cultures and natural communities are now needed to better constrain the role of fungi in carbon processing in the present and in response to climate change. Understanding the biological processes of oceanic systems is crucial to study the influence of climate change [2], and pelagic fungi cannot be neglected from the processes governing the marine carbon cycle anymore. In the future, further research should focus on increasing the knowledge about respiration, production, and growth efficiency of marine pelagic fungi.

## Figures and Tables

**Figure 1 jof-09-00417-f001:**
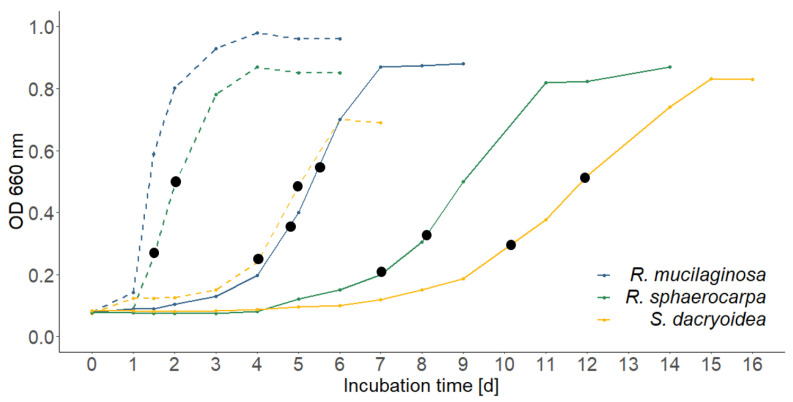
Growth curves of *Rhodotorula mucilaginosa*, *Rhodotorula sphaerocarpa*, and *Sakaguchia dacryoidea* in the high nutrient media; measured via optical density (OD; 660 nm). Dashed lines represent the growth at 15 °C and continuous lines the growth at 5 °C. The black dots indicate the sampling points during the exponential growth phase.

**Figure 2 jof-09-00417-f002:**
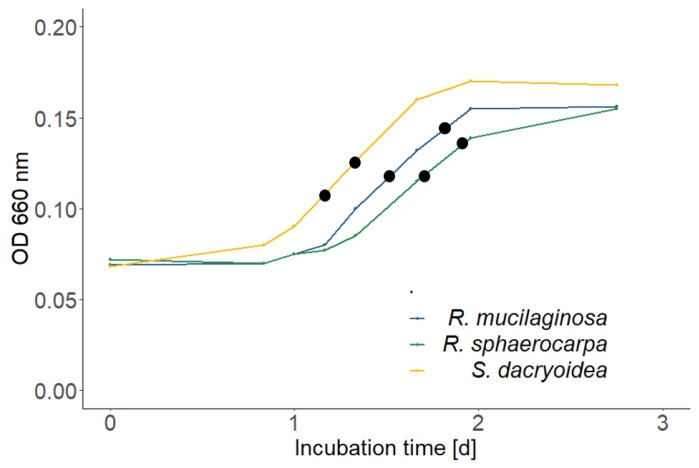
Growth curves of *Rhodotorula mucilaginosa*, *Rhodotorula sphaerocarpa*, and *Sakaguchia dacryoidea* in the low nutrient media; measured via optical density (OD; 660 nm). The black dots indicate the sampling points during the exponential growth phase.

**Figure 3 jof-09-00417-f003:**
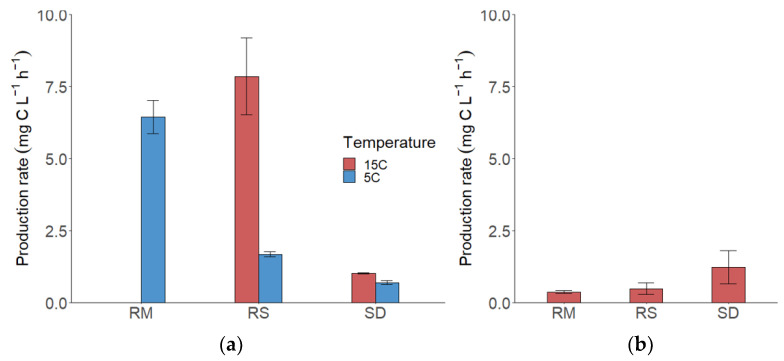
Production rates of the three fungal isolates *Rhodotorula mucilaginosa* (RM), *Rhodotorula sphaerocarpa* (RS), and *Sakaguchia dacryoidea* (SD), bars represent the mean production, and the error bars the standard error (SE). (**a**) Production rates in the high nutrient media at 5 °C (blue bars) and 15 °C (red bars; due to technical issues no measurements were performed in the *R. mucilaginosa* culture). (**b**) Production rates in the low nutrient media at 15 °C.

**Figure 4 jof-09-00417-f004:**
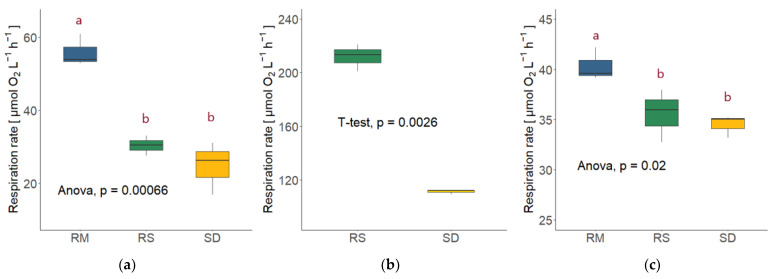
Respiration rates of the three fungal isolates *Rhodotorula mucilaginosa* (RM), *Rhodotorula sphaerocarpa* (RS), and *Sakaguchia dacryoidea* (SD) with the respective statistical test (ANOVA or Student’s *t*-test). Red letters indicate which rates differed significantly from each other, as determined by a TukeyHSD post-hoc test. (**a**) Respiration rates in the high nutrient media at 5 °C. (**b**) Respiration rates in the high nutrient media at 15 °C (due to technical issues no measurements were performed in the *R. mucilaginosa* culture). (**c**) Respiration rates in the low nutrient media at 15 °C.

**Figure 5 jof-09-00417-f005:**
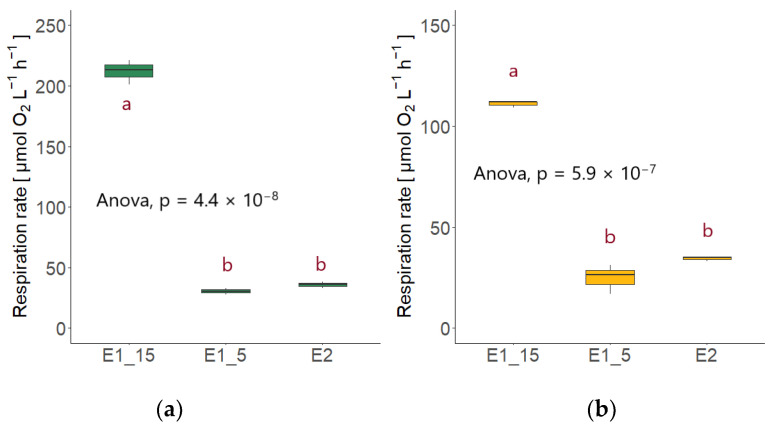
Respiration rates of the fungal cultures in the high nutrient media at 15 °C (E1_15) and at 5 °C (E1_5) and in the low nutrient media (E2) with the respective statistical test (ANOVA). Red letters indicate which treatments differed significantly from each other, as determined by a TukeyHSD post-hoc test. (**a**) Respiration rates of *Rhodotorula sphaerocarpa*. (**b**) Respiration rates of *Sakaguchia dacryoidea*.

**Figure 6 jof-09-00417-f006:**
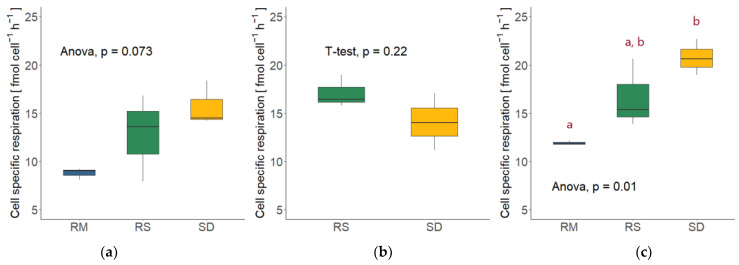
Cell-specific respiration rates of the three fungal isolates *Rhodotorula mucilaginosa* (RM), *Rhodotorula sphaerocarpa* (RS), and *Sakaguchia dacryoidea* (SD) with the respective statistical test. Red letters indicate which rates differed significantly from each other, as determined by a TukeyHSD post-hoc test. (**a**) Cell-specific respiration rates in the high nutrient media at 5 °C. (**b**) Cell-specific respiration in the high nutrient media at 15 °C (due to technical issues no measurements were performed in the *R. mucilaginosa* culture). (**c**) Cell-specific respiration rates in the low nutrient media at 15 °C.

**Figure 7 jof-09-00417-f007:**
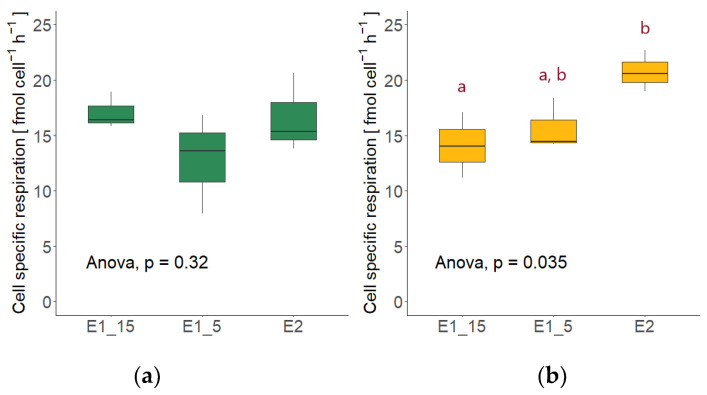
Cell-specific respiration rates of the fungal cultures in the high nutrient media at 15 °C (E1_15) and at 5 °C (E1_5) and in the low nutrient media (E2) with the respective statistical test. Red letters indicate which treatments differed significantly from each other, as determined by a TukeyHSD post-hoc test. (**a**) Respiration rates of *Rhodotorula sphaerocarpa*. (**b**) Respiration rates of *Sakaguchia dacryoidea*.

**Table 1 jof-09-00417-t001:** Ingredients of the media for the agar plates and liquid cultures in g/L.

Compounds [g/L]	Agar Plates	High Nutrient Media	Low Nutrient Media
D-(+)-Glucose (Sigma Aldrich)	10	0.5	0.05
Malt extract (Merck KGaA)	5	-	-
Yeast extract (Sigma Aldrich)	3	0.5	0.05
Bacto^TM^ Peptone (BD)	5	0.5	0.05
Agar (Sigma Aldrich)	20	-	-
Sea Salts (Sigma Aldrich)	30	30	30
Chloramphenicol (Sigma Aldrich)	0.5	0.5	0.5

**Table 2 jof-09-00417-t002:** Optical density (OD) values measured at 660 nm for estimating the growth phase of the liquid cultures in the high and low nutrient media.

Estimated Growth Phase	OD in the High Nutrient Media	OD in the Low Nutrient Media
Lag phase	<0.2	<0.1
Exponential phase	0.2–0.6	0.1–0.154
Stationary phase	>0.6	>0.154

**Table 3 jof-09-00417-t003:** Optical density (OD) of the three species *Rhodotorula mucilaginosa*, *Rhodotorula sphaerocarpa*, and *Sakaguchia dacryoidea* at the first and second sampling point (SP; see Figure 1 and Figure 2 for reference) in the high and low nutrient media. Numbers are given in average OD and the numbers in brackets indicate the standard deviation.

Species	High Nutrient Media				Low Nutrient Media	
	15 °C		5 °C			
	SP1	SP2	SP1	SP2	SP1	SP2
*R. mucilaginosa*			0.31 (0.02)	0.52 (0.04)	0.13 (0.002)	0.15 (0.002)
*R. sphaerocarpa*	0.29 (0.01)	0.47 (0.02)	0.201 (0.001)	0.29 (0.04)	0.12 (0.004)	0.14 (0.002)
*S. dacryoidea*	0.23 (0.01)	0.47 (0.004)	0.25 (0.03)	0.55 (0.03)	0.109 (0.002)	0.13 (0.002)

**Table 4 jof-09-00417-t004:** Q_10_ values of the total respiration, cell-specific respiration, and production rates of *Rhodotorula sphaerocarpa* and *Sakaguchia dacryoidea*. The values derive from comparing the measurements at 5 °C and 15 °C in the high nutrient media.

Species	Total Respiration	Cell-Specific Respiration	Production
*Rhodotorula sphaerocarpa*	4.49	1.33	4.68
*Sakaguchia dacryoidea*	2.14	0.89	1.46

**Table 5 jof-09-00417-t005:** Cell abundance of the three species *Rhodotorula mucilaginosa*, *Rhodotorula sphaerocarpa*, and *Sakaguchia dacryoidea* at the first and second sampling point (SP; see Figure 1 and Figure 2 for reference) in the high and low nutrient media. Numbers are given in average cells L^−1^ and the numbers in brackets indicate the respective standard deviation.

Species	High Nutrient Media				Low Nutrient Media	
	15 °C		5 °C			
	SP1	SP2	SP1	SP2	SP1	SP2
*R. mucilaginosa*			1.07 × 10^10^ (1.28 × 10^9^)	1.50 × 10^10^ (1.9 × 10^9^)	3.39 × 10^9^ (8.53 × 10^7^)	3.75 × 10^9^ (2.18 × 10^8^)
*R. sphaerocarpa*	7.16 × 10^9^ (9.63 × 10^8^)	1.25 × 10^10^ (7.13 × 10^8^)	1.92 × 10^9^ (3.49 × 10^8^)	2.61 × 10^9^ (1.06 × 10^9^)	2.97 × 10^9^ (1.12 × 10^8^)	2.50 × 10^9^ (3.60 × 10^8^)
*S. dacryoidea*	1.18 × 10^9^ (2.14 × 10^8^)	8.39 × 10^9^ (1.47 × 10^9^)	8.85 × 10^8^ (5.22 × 10^8^)	3.08 × 10^9^ (6.06 × 10^8^)	1.67 × 10^9^ (1.98 × 10^8^)	3.01 × 10^9^ (1.16 × 10^8^)

**Table 6 jof-09-00417-t006:** Review table of examples of studies in which growth efficiency of cultures of marine organisms were estimated.

Organism(s)	Culture Conditions	Growth Efficiency	Reference
Bacterial community of the Bothnian Sea	Batch culture with vs. without nutrients addition	14–58%	[42]
		11–54%	
Bacterial community from the Gulf of Mexico	Batch culture	61%	[43]
Bacterial community of the coast off Massachusetts	Batch culture vs. Continuous culture	34–70%	[44]
		43–58%	
Marine bacterial isolate (*Vibiro harveyi*)	Pure culture with added glucose and iron	45–55%	[45]
Bacterial community of the Mediterranean	Seawater cultures with added phosphorus	17–70%	[33]
Marine bacterial isolates (*Roseobacter* and *Cytophaga*)	Pure cultures in seawater and added Zobell media	50–70%	[34]
Marine bacterial isolates (*Vibrio splendidus* and *Phaeobacter gallaeciensis*)	Batch cultures with glucose as carbon source	9–25%	[46]
Marine fungal isolates (Basidiomycota)	Pure cultures in defined growth media	43–87%	This study

## Data Availability

The raw data supporting the conclusions of this article will be made available by the authors, without undue reservation to any qualified researcher.

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
