# Peer review of "Respiration, Production, and Growth Efficiency of Marine Pelagic Fungal Isolates"

_jof, 2023, doi:10.3390/jof9040417_

Round 1

Reviewer 1 Report

The authors conducted a seemingly simple but very interesting study related to growth, respiration and production of marine fungi under artificial environment, which highlights their importance in carbon cycle in the ocean. The manuscript was written well, and the results were explained clearly. One issue I concerned is that some descriptions are arbitrary. For example, “this study is a first attempt to investigate respiration and production rates, and the growth efficiency of marine pelagic fungi”. I disagree this conclusion. In fact, many researchers in different countries have done similar studies, but they published their data in different languages such as Japanese, Chinese etc. Table 5 faces the same question. It would be better if the authors take these studies listed in Table 5 as some typical examples.  

Reviewer 2 Report

In the manuscript submitted by Heitger & Baltar, three yeast isolates from the Basidiomycota are used to quantify some important properties that can be used to quantitatively assess the role of pelagic marine fungi in the context of the carbon cycle. An important parameter is the respiration rate and the growth efficiency under different nutrient regimes as well as two temperatures.

The idea of the approach is good and the results are discussed very well. For example, they are compared with known values of pelagic bacteria and interpreted what this might mean for the ecological role of marine fungi in carbon turnover.

Unfortunately, the material and methods section is written in a rather intransparent way. Important information is missing, scattered or redundant. In other places, it would be helpful to get much more information. Therefore, as a reader, the exact approach is unclear to me in several points. Unfortunately, I could not understand how many replicates were used for the individual measurements. There is also no SD for the growth curves, which unfortunately calls the quality of the experiment into question. What are the controls? E.g. do the fungi grow out of their internal storage and does this affect the growth curves? I would encourage repeating individual experiments with controls and replicates where necessary. The message of this paper is so interesting that I believe it is worthwhile.

Furthermore, I suggest the authors to stringently revise the MM part, which will significantly improve the quality of the manuscript. I have given many details below and hope that this will help the authors to understand my criticism.

13: how many nutrient concentrations? Which type of nutrients?

15: production versus growth efficiency?

38-39: place in paragraph above

48: no, shows only that there is more biomass of fungi compared to bacteria

56-59: cite Orsi et al. 2022 (10.1038/s41396-021-01169-5) and Chrismas et al. 2020 (10.1038/s41396-020-0687-2).

67: why attempt?

68-77: shift into MM

83-86: which media? Which nutrients do you talk about? Which salinity? Please, be very precise as this is the basis of your experiment.

89: how?

Table 1: provide company and detailed information on chemicals. What do you mean by sea salt???

91: no antibiotics?

96: 30 of what? Unit is missing

97: when have you used which seawater/sea salt?

99-100: What is the relationship between the used conc and natural conc?

103: what is the inoculation step? What is the amount of cells or biomass used for the inoculation? Do you have the same biomass in the different 200 ml? what are positive and what are negative controls? How many replicates?

104: which incubator exactly?

107: you did not measure OD from all replicates/bottles? Why?

108: provide data from pre-experiment

110: from how many bottles have you taken the values? What was the OD of the two timepoints?

121: what you are measuring is the total organic carbon (TOC)-content of the biomass

125: on which instrument? Did you stick only to the protocol given in 24?

127: have you done this for each measurement?

131: time as what?

141: how have you calibrated the instrument on yeasts? What was the size of the cells?

150: what does “close to” mean?

144-164: this was done independently of the other measurements? Which conditions did you use? What was the OD or cell numbers for the start of the incubations?

162-164: Please provide some background information on the conversion into carbon units. The approach is not comprehensible.

168-169: Q10: provide more information

176: which parameters?

178-182: This is written very vague. Under what P-values were the individual tests performed? What are the groups being tested and what questions were addressed?

Fig 1 & 2: No standard deviations are shown. Why were no growth curves measured at 5°C in low nutrient medium?

246-247: Why can't the values of e.g. the respiration rate be converted to cell counts instead of "similar" OD, which can introduce a quite strong error.

Figure 3: can you perform statistics on the production rate?

Fig 4: The text says that Tukey was used to detect differences in respiration rates (please correct the text, it does not compare species) of individual species. But in the figure itself there are other tests. This is very confusing. Perhaps this could be better presented graphically.

346-354: I find this idea very important. It answers so many questions: why do we have the great diversity of species? Why is there so much dynamic in mycopelagic communities? It would be best if this idea were already in the introduction. In my opinion, it would give the paper a completely different weight in terms of the ecological role of marine fungi.

391: Is this a value for the pelagic microbial community or just bacteria?

Table 5: What does„continuous“culture mean?

Round 2

Reviewer 2 Report

two minor points:

32: it is not the respiration but the growth efficiency (simply because respiration leads to CO2 production and thus carbon goes back into the atmosphere)

113: PSU?

Very interesting study!